# A Water Surface Contaminants Monitoring Method Based on Airborne Depth Reasoning

**Wei Luo** [1,2,3], **Wenlong Han** [1], **Ping Fu** [4], **Huijuan Wang** [1], **Yunfeng Zhao** [1,2,3], **Ke Liu** [1,2,3], **Yuyan Liu** [1,2,3], **Zihui Zhao** [1,2,3,5,*], **Mengxu Zhu** [1], **Ruopeng Xu** [1] **and Guosheng Wei** [1]

[1] North China Institute of Aerospace Engineering, Langfang 065000, China; luowei@radi.ac.cn (W.L.); m17803269306@163.com (W.H.); wanghj@nciae.edu.cn (H.W.); nanhchao@163.com (Y.Z.); liuke1176@163.com (K.L.); lyy@nciae.ac.cn (Y.L.); mengxu_zhu@163.com (M.Z.); sam970125@163.com (R.X.); convwei@163.com (G.W.)

[2] Aerospace Remote Sensing Information Processing and Application Collaborative Innovation Center of Hebei Province, Langfang 065000, China

[3] National Joint Engineering Research Center of Space Remote Sensing Information Application Technology, Langfang 065000, China

[4] Key Laboratory of Advanced Motion Control, Fujian Provincial Education Department, Minjiang University, Fuzhou 350108, China; shirkl@163.com

[5] School of Geography, Hebei Normal University, Shijiazhuang 050024, China

[*] Correspondence: 13770765697@163.com

**Abstract:** Water surface plastic pollution turns out to be a global issue, having aroused rising attention worldwide. How to monitor water surface plastic waste in real time and accurately collect and analyze the relevant numerical data has become a hotspot in water environment research. (1) Background: Over the past few years, unmanned aerial vehicles (UAVs) have been progressively adopted to conduct studies on the monitoring of water surface plastic waste. On the whole, the monitored data are stored in the UAVS to be subsequently retrieved and analyzed, thereby probably causing the loss of real-time information and hindering the whole monitoring process from being fully automated. (2) Methods: An investigation was conducted on the relationship, function and relevant mechanism between various types of plastic waste in the water surface system. On that basis, this study built a deep learning-based lightweight water surface plastic waste detection model, which was capable of automatically detecting and locating different water surface plastic waste. Moreover, a UAV platform-based edge computing architecture was built. (3) Results: The delay of return task data and UAV energy consumption were effectively reduced, and computing and network resources were optimally allocated. (4) Conclusions: The UAV platform based on airborne depth reasoning is expected to be the mainstream means of water environment monitoring in the future.

**Keywords:** deep learning; edge computing; machine learning; open source unmanned aerial vehicle; plastic waste detection; remote sensing; water environment protection

## 1. Introduction

Plastic refers to a type of high polymer compound that is characterized by differing compositions and shape flexibility. It exhibits several advantages (e.g., impact resistance, wear resistance, good insulation and low cost), nonetheless, it has significant defects. Additionally, plastic cannot be effectively recovered through classification. As indicated from the report of the voice of economy, the industrial ecology team of the University of California estimated the overall amount of plastics available on Earth. Since it was invented in 1909, mankind has produced approximately nine billion tons of plastics, equated with 25,000 Empire State buildings in New York and the sum of one billion elephants (e.g., plastic bottles, plastic bags and other plastic products). Merely 30% of plastic is recycled, and 70% of plastic turns out to be garbage, most of which is buried under the land.

According to the team of California scientists, considerable garbage floats in the ocean. Plastic is suggested to be not biodegradable. Over time, macro plastic pieces degrade into increasingly smaller pieces, termed microplastic (less than five millimeters long) [1]. Microplastic can be swallowed by various water surface organisms and then rise through the food chain, ending up on our dinner tables [2]. Water surface plastic waste pollution is a major challenge to the global ecology and impacts numerous fields (e.g., economics, ecology, public health and aesthetics).

The amount of plastic waste in the ocean has reached 150 million tons, nearly one fifth of the total weight of marine fish [3]. It is estimated that the weight of plastic in the ocean will exceed that of fish by 2050. The international community has made some efforts to build a standardized monitoring method, including Oslo and Paris Conventions (OSPAR) (OSPAR commission, 2020) [4], Commonwealth Scientific and Industrial Research Organization (CSIRO) [5], National Oceanic and Atmospheric Administration (NOAA) [6], as well as United Nations Environment Programme/Intergovernmental Oceanographic Commission (UNEP/IOC) [7]. However, little knowledge has been acquired from the total quantity and spatial-temporal distribution of water surface plastic waste, and the monitoring method remains in the preliminary stage. Mauro et al. [8] inserted 190 FTIR spectra of plastic samples in a digital database and submitted those to Independent Component Analysis (ICA) to extract the "pure" plastic polymers present. In addition, they established the similarity with unknown plastics by employing the correlation coefficient (r), and the cross-correlation function (CC). Topouzelis K. et al. [9] adopted worldview-2 images to examine the optical properties exhibited by wet and dry plastics, as well as assessed the possibility of multispectral images for floating plastic detection in water. Kyriacos et al. [10] set seven indices for satellite image processing, which were examined to verify whether they are capable of detecting plastic waste in water. Furthermore, the authors examined two novel indices to be applied for processing satellite images, i.e., the Plastics Index (PI) and the Reversed Normalized Difference Vegetation Index (RNDVI). The novel Plastic Index (PI) is capable of detecting plastic objects floating on the water surface, and it has been proven as the most effective index to detect the plastic waste target in the sea. By mounting the equipment on a C-130 aircraft that surveyed the Great Pacific Garbage Patch, Shungudzemwoyo et al. [11] captured red, green and blue (RGB) and hyperspectral SWIR imagery. Furthermore, they explored SWIR spectral information acquired by employing a SASI-600 imager (950−2450 nm) and then examined the potential of SWIR remote sensing technology in detecting and quantifying ocean plastic.

Unmanned Aerial Vehicles (UAVs) have been demonstrated as an effective low-cost image-capturing platform capable of accurately monitoring aquatic environments [12,13]. Gil et al. [14] proposed an Unmanned Aerial System (UAS)-based process for automated water surface litter mapping under a beach-dune system. The very high-resolution orthophoto produced from UAS images was automatically screened by the random forest machine learning method to characterize the water surface litter load on beach and dune areas. A. Deidun et al. [15] optimized the protocol to monitor the identical litter along coastal stretches within an MPA in the Maltese Islands through aerial drones, with the aim of generating density maps for the beached litter. The mentioned process can help detect the identical litter and mainstream such a methodology in national and regional programs for monitoring water surface waste. UAVs have been exploited to capture geo-referenced RGB images in the selected zone of a protected water surface area (the Migliarino, Massacciuccoli and San Rossore Park near Pisa, Italy) in a long-term (ten month) monitoring program. A post-processing system based on visual interpretation of the images can be applicable to localizing and detecting the anthropogenic water surface debris in the scanned area, as well as estimating their spatial and temporal distributions in different beach zones [16].

Over the past few years, deep learning theory and the practice of theory have been trailblazing, and the theory has been applied for detecting water surface plastic waste. The deep learning model can automatically select image features, which is considered an advantage of the model. VGGNet [17], FCN [18], Faster R-CNN [19], Yolo [20], U-Net [21]

and other models exhibit the most advanced accuracy in detecting floating plastic waste in UAV images. Kyriaki et al. [22] proposed a macro plastic recognition model by complying with the convolutional neural network (CNN). When the classifier is trained on three identical types of plastic water surface garbage (i.e., plastic bottles, plastic buckets and plastic straws), it is capable of recognizing novel plastic objects well, and the verification accuracy reaches approximately 86%. Jun Ichiro et al. [23] explored the method of exploiting autonomous robots (e.g., commercial UAVs and AUVs) to monitor water surface environments. Moreover, they adopted the deep learning target detection algorithm Yolov3 to detect underwater water surface organisms and floating debris on the sea, achieving the respective average accuracies of 69.6% and 77.2%. However, they ignored the top-level spatial information, thereby causing the lack of accurate positioning and class boundary characterization. Furthermore, the mentioned methods are primarily offline analysis methods based on aerial photography data acquired by using UAVs, i.e., UAVs acquire real-time information from the surrounding environment by turning on the camera or sensor while synchronously pushing it to the ground station. Subsequently, the ground station transmits the video or image information acquired to the special image analysis server (workstation) for subsequent analysis. All the above process is considered a significant resource-intensive task. The model network is sophisticated with considerable parameters and low efficiency, and the degree of real-time is largely determined by the bandwidth and stability of the transmission network.

The strategy of edge computing can address the mentioned problems. Edge computing decomposes the large-scale services originally processed by the central node and disperses them to the edge nodes closer to the user terminal equipment. It is capable of expediting the data processing and sending and reducing the delay. Kang Z. et al. [24] used the flight points calculated by UAV to fly in turn to cover a convex polygon area. The detailed solving process of flight point was given, while the programming, simulation and actual flight experiment of the proposed method were performed. Zhang Z. et al. [25] first used the deep learning model to preprocess the captured image and extracts useful information. Subsequently, they transmitted these data to the edge server on the ground for further analysis. Compared with the direct transmission of the original data, this method is capable of significantly reducing the communication load.

The existing airborne image processing board has limited computing power, so it cannot easily perform large-scale target solving tasks. As a typical one-stage algorithm, Yolo series of target detection algorithms exhibit high precision and are fast and lightweight. According to the latest yolov5, the fast reasoning time of the respective image is up to 0.007 s and 140 frames per second (FPS). Yolov5, an extremely lightweight target recognition network, solves the problems of low efficiency of the full convolution model network, as well as the difficulty in ensuring the classification effect. As indicated from the verification of several public datasets, its accuracy is equivalent to that of EfficientDet and yolov4, whereas the model size is only one tenth of the latter [26,27]. It is an ideal choice to carry out edge computing on UAVs, unmanned ships and other platforms.

The rest of this study is organized below. First, in the second section, the research area of the water surface plastic waste monitoring experiment is introduced, as well as the photoelectric pod and target detection model applied by the aerial robot. Subsequently, in the third section, the computer configuration of Aerial Robot and the training method of target detection model is presented. To solve the problems of River waste monitoring, this study proposed three optimization strategies, compared and analyzed the models and discussed the recognition results of different models. Lastly, in the fourth section, the specific challenges and future development trend of Aerial Robot are summarized for real-time monitoring of plastic waste on the water surface.

## 2. Materials and Methods

### 2.1. Study Area and UAV Trajectory Planning

The experiment of this study was conducted at the East Zhangwu Wetland Section of Longhe river in Anci District, Langfang City, Hebei Province. The longitude amplitude of the research area ranged from 116.69° N to 116.80° N, while the latitude amplitude was from 39.41° E to 39.48° E. Longhe River refers to an interprovincial and intermunicipal drainage channel, originating from Daxing District of Beijing and entering Yongding River flooding area via a dike protection road in East Zhangwu of Langfang City. The river exhibits a total length of 68.42 km and a drainage area of 577.94 km$^2$, of which 256 km$^2$ is in Beijing and 322 km$^2$ is in Langfang. The Longhe river serves as a vital barrier to protect the ecological balance of the capital city.

The main water surface environmental protection method in East Zhangwu Wetland section of Longhe River aims at irregular manual inspections and fishing as assisted by a diesel-powered fishing boat (Figure 1). As impacted by the long river length, there are often people picnicking and camping along its shores, so the possibility of sudden plastic waste pollution is high. The existing salvage vessels have slow speeds (<20 km/h), the number of operators is small (2 people) and the emergency response ability to sudden pollution is low. Furthermore, if the number of inspections is extremely frequent, the exhaust gas emitted by the vessels causes secondary pollution to the wetland environment.

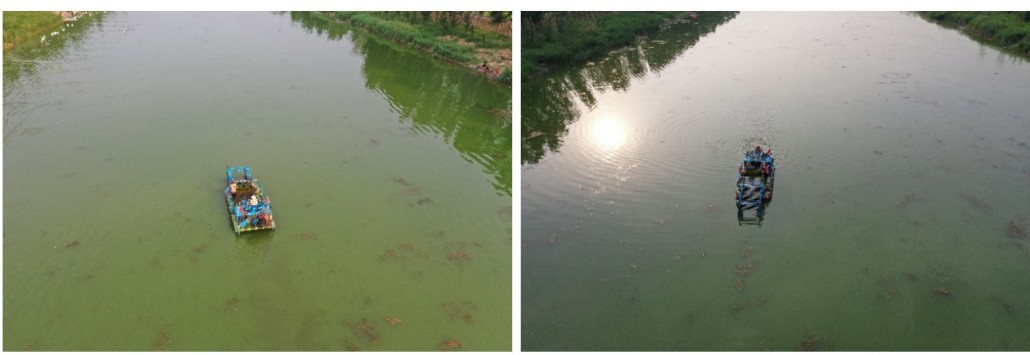

**Figure 1.** Salvage vessel for water surface waste inspection on the Longhe River.

The aerial robot can substitute for the salvage vessel to irregularly inspect the river environment and send the waste detection results to the ground station on the salvage vessel in real time. When the waste loading [28] reaches a certain degree, the salvage vessel can perform accurate salvage in accordance with the waste position fed back by the aerial robot. It is capable of significantly increasing the efficiency of salvage and saving capital and labor costs while reducing the exhaust pollution attributed to repeated vessel inspections.

QGroundControl software was adopted to design the flight route of the survey area. QGroundControl can offer full flight control and the vehicle setup for PX4 or ArduPilot-powered vehicles. To yield the optimal resolution, numerous experiments were performed, and the optimal altitude of 7 m was determined. The speed was set at 5 m/s, the course overlap rate was 80% and the side overlap rate was 75%. The flight route was perpendicular to the river flow direction (Figure 2).

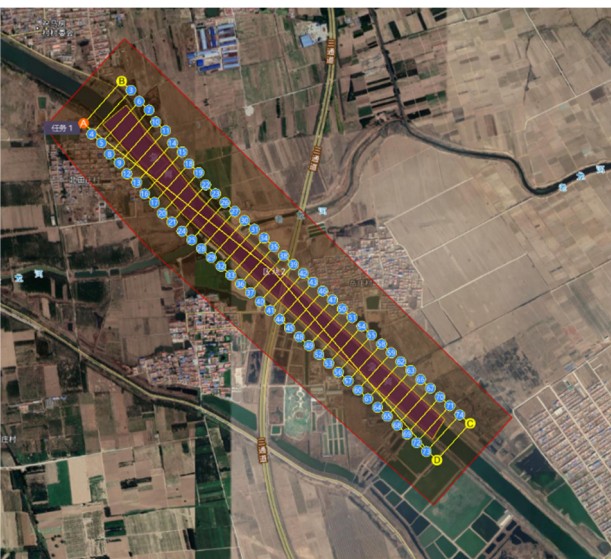

**Figure 2.** Distribution of survey area.

### 2.2. Data Acquisition

The remote sensing images of the study area were captured with an Intel d435i binocular depth camera on a Feisi x450 UAV (Figure 3a) developed by Beijing Droneyee Intelligent Technology Co., Ltd., Beijing, China. There were four round holes on the front of the d435 camera (Figure 3b). From left to right, the first and third were IR stereo cameras, while the second and the fourth were an IR projector and a color camera, respectively. The maximal distance of camera capture was 10 m, and the video transmission rate could be up to 90 FPS. Feisi x450. The UAV is also equipped with a TX2 airborne visual processing board, capable of performing visual navigation, target recognition and target following. Other tasks will be introduced in Section 3.1.

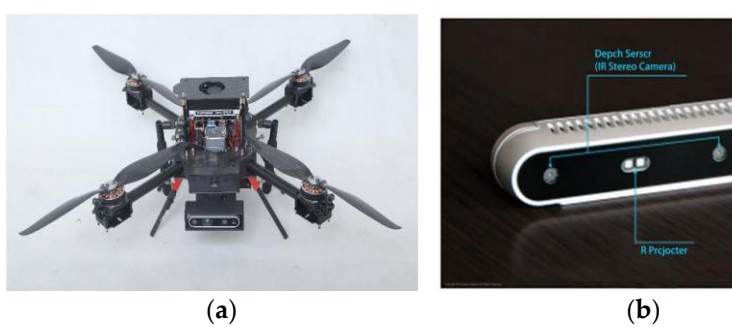

|         (a)          |          (b)          |

**Figure 3.** Data acquisition equipment of the study. (**a**) FEISI X450 UAV; (**b**) Intel d435i binocular camera.

From the UAV survey, 10,000 UAV remote sensing images were selected area as the sample database, with an image resolution of about 1 cm and a size of 2048 × 1080 pixels. Of the 10,000 images, 7000 were employed as the training samples, and the remaining 3000 were employed as test samples to verify the recognition results. The data of this research are available in ScienceDatabank (doi:10.11922/sciencedb.01121). This study used professional labeling software in the UAV images to mark common plastic waste (e.g., plastic bottles, plastic bags and plastic foam). The coordinates of the upper left corner and the lower right corner of the rectangle box were recorded in an XML document.

As impacted by the small amount of plastic waste in the UAV remote sensing image, to obtain higher training effect, the data were partially downloaded from the public dataset as a supplement. Moreover, the data were randomly cut, rotated, scaled and flipped

to generate multiple similar images. Data enhancement is capable of compensating for incomplete data, effectively reducing the overfitting, making the model more applicable to novel samples and improving the generalization utility of the model. Lastly, UAV images were converted into datasets of visual object class format for pretraining of the deep learning model.

### 2.3. Overall Research Framework

The overall research framework of this study is shown in Figure 4. First, the model was pretrained by the open dataset. Subsequently, the model was trained and reasoned by the labeled (interpretation object) training set. Given the characteristics exhibited by plastic waste, the deep learning model was regulated to achieve a more effective solution and then packaged. By building the intelligent analysis platform of edge computing UAV, the encapsulated deep learning model was transplanted to the airborne image processing board.

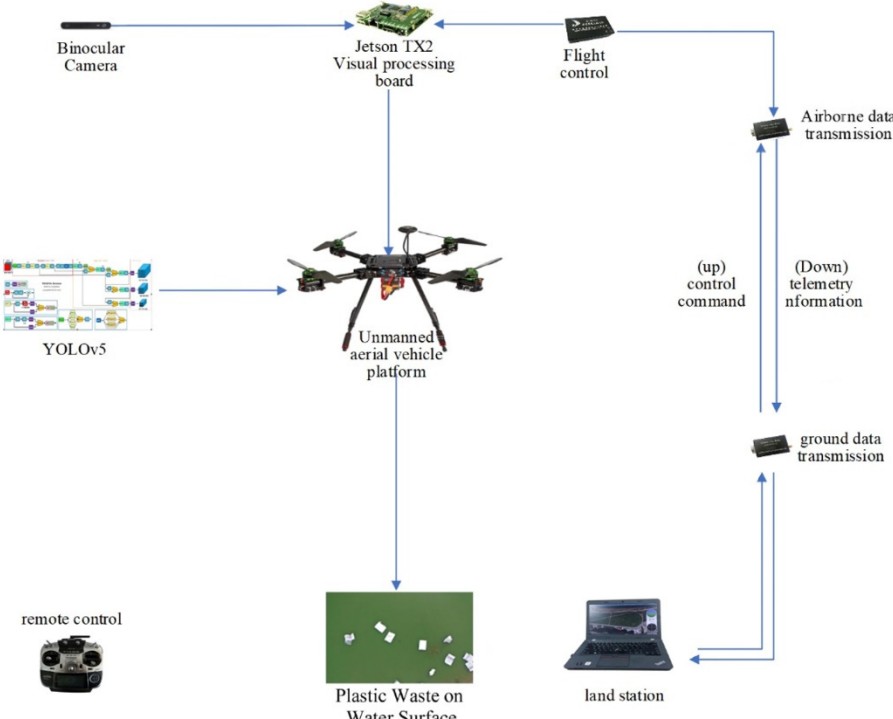

**Figure 4.** Research framework of this study.

The UAV processed and analyzed the plastic waste targets while capturing aerial photos. In addition, it detected and counted the types and quantities of plastic waste while transmitting the detection results back to the ground workstation on the salvage vessel via data transmission. The operator decided when to perform the fishing work in accordance with the quantity and position of plastic waste on the water surface obtained by the workstation in real time.

### 2.4. Deep Network

The target detection of a flowing river is significantly challenging. There are many challenges attributed to continuous plastic movement (e.g., low amount of training data, high imbalance of dataset, frequent target location and scene changes). This study took yolov5 as the baseline algorithm and proposed various optimization strategies to address the problems in target detection. The overall flow chart of the algorithm is presented in Figure 5.

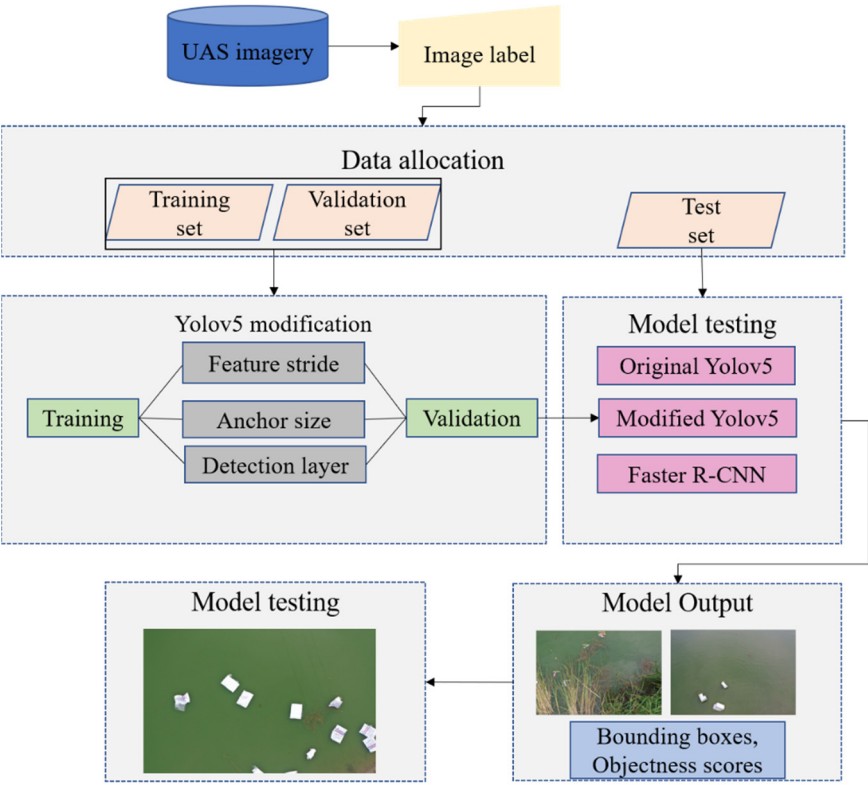

**Figure 5.** Algorithm block diagram.

In this study, a novel deep learning network for target detection, yolov5, was adopted to detect the labeled training set image. Yolov5 refers to the network exhibiting the smallest depth and feature map width in the target detection series, with its accuracy equivalent to that of yolov4, whereas the model is nearly 90% smaller than that of yolov4. Yolov5 is considered a prominent lightweight network with fast convergence on multiple datasets and high customizability. The relevant source code can be referenced from https://github.com/ultralytics/YOLOv5 accessed on 9 April 2021. Yolov5 was implemented by complying with the Python framework. Yolov5 operations place a novel focus on changing the image into a feature map after slicing. Two CSP structures were applied in the backbone extraction network. With yolov5's network as an example, a csp1_ X structure was applied to the backbone network, i.e., another type of csp2. The X structure was used in the neck. The FPN + pan structure was selected as the neck. The csp2 structure designed by cspnet was employed to improve the ability of network feature fusion.

CSP structure divides the feature map into two parts, and then merges it through the proposed cross stage hierarchy. By splitting the gradient flow, the gradient flow propagates through different network paths. It can greatly reduce the amount of calculation and improve the reasoning speed and accuracy. Two CSP structures, csp1, are used in yolov5 network_ X for backbone feature extraction network, csp1_ X uses the residual structure module to speed up the backbone feature extraction and network feature extraction capability. CSP2_ X is used for the neck network, FPN + pan structure is selected as the neck, and csp2 is used_ X structure to improve the ability of network feature fusion.

Yolo, a highly typical target detection algorithm, refers to a single-stage algorithm integrating target proposal stage and classification stage, and its detection rate is higher than that of the two-stage RNN algorithm. Yolov5 is regarded as the latest version of the Yolo architecture. Yolov5 architecture comprises four architectures, i.e., Yolov5s, Yolov5m, Yolov5l and Yolov5x. To prevent the model from being extreme and overfitting, this study selected Yolov5s with a relatively simple structure as the baseline model.

On the whole, Yolov5s framework comprises three parts (i.e., backbone network, neck network and detection network). The backbone network aims to aggregate different

convolutional neural network images, as an attempt to form image features. To be specific, the first layer of the backbone network is the focus module. First, the respective input UAV image fell to four slices, and the slice operation was used to reduce the amount of model calculation and increase the training speed without image loss. Second, the four parts were deeply connected by concat operation to output the size of the characteristic graph. Subsequently, the results were outputted to the next layer via the convolution layer (conv2d + BN + leakyrelu activation function, CBL) composed of 32 convolution cores.

The third layer of the backbone network refers to the BottleNeckCSP module used to more effectively extract the deep features of the image. The Bottleneck CSP module primarily consists of the bottleneck module. It connects the $1 \times 1$ CBL and $3 \times 3$ residual network architecture of CBL. The ninth layer of the backbone network is the SPP module (spatial pyramid pooling), converting any size feature map into a fixed size feature vector to optimize the receptive field of the network. First, the neck network is the feature map output after the convolution layer. The feature map is linked to the sub sampling depth of the output feature map via three parallel maxpooling layers. The output feature map is capable of retrieving the final output feature map via a convolution layer.

The neck network, a series of hybrid feature aggregation layer image features, is largely exploited to generate a feature pyramid network and subsequently transmit the output feature map to the detection network. The feature pyramid network structure optimizes the bottom-up path, improves the transmission of low-level features and facilitates the detection of floating plastic waste at different scales. Thus, the same target object with different sizes and scales can be accurately detected. The detection network was primarily applied for the last detection part of the model. The anchor box was applied into the feature map output on the neck network, and a vector was outputted (e.g., the category probability of the target object, the score of the object and the position of the bounding box around the object). The detection network of Yolov5s architecture comprises three detection layers, which are adopted to detect image objects of different sizes. Lastly, the respective detection layer outputted a vector while generating and marking the prediction boundary box and category of the target in the original image to detect plastic waste in the UAV image.

To increase the accuracy of water surface garbage detection, the original model was converted to the modified Yolov5:

1.　Modifying anchor structure

Anchor structure, a vital part of the Yolo series target detection algorithm, produces suggestions for predicting potential objects. The original anchor structure exhibits high performance in detecting various objects in the dataset (e.g., coco). However, the size of these anchors is not applicable to small objects. The average size of plastic waste in the UAV image here was less than 30 cm, and the total area of the image was only approximately 1% of the overall image area. In small target detection, setting a small anchor scale is considered a feasible solution to solve the mentioned problem. However, it is arbitrary to assess the performance of the model by comparing the anchor size and sample size, and the model is also capable of finding a more appropriate size by the bounding box regression. To select the appropriate anchor size, the anchor size selection setting was optimized by a K-means clustering algorithm and then set experimentally [45,62; 25,20; 16,28], [13,9; 31,44; 10,26], [24,54; 15,21; 23,30]. Three groups of anchor structures were tested for the target (Figure 6).

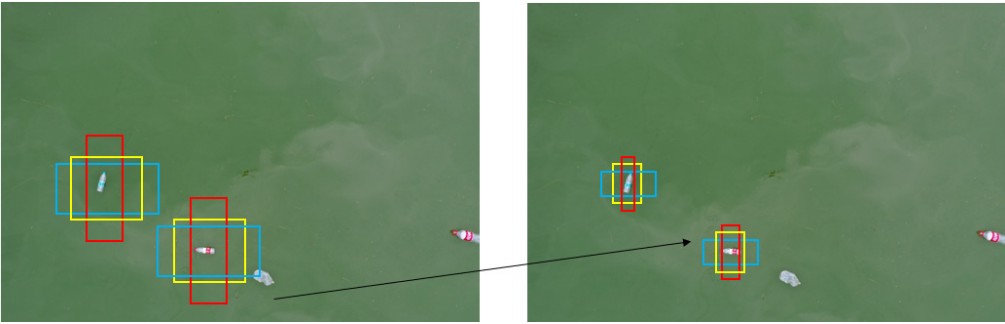

**Figure 6.** Modification of anchor structure.

2.     Modify step size

Between the convolution layer and the aggregation layer of deep learning in the network, several layers have steps larger than 1, thereby performing the down-sampling operation and generating a series of smaller feature maps. The category and location of small objects are difficult to predict in large step down-sampling. A simple and effective method can be used to reduce the feature step for tackling down the down-sampling problem in the small target detection. The 16strides was modified to 8strides to make the feature extraction network accurately extract the plastic garbage in the UAV network (Figure 7).

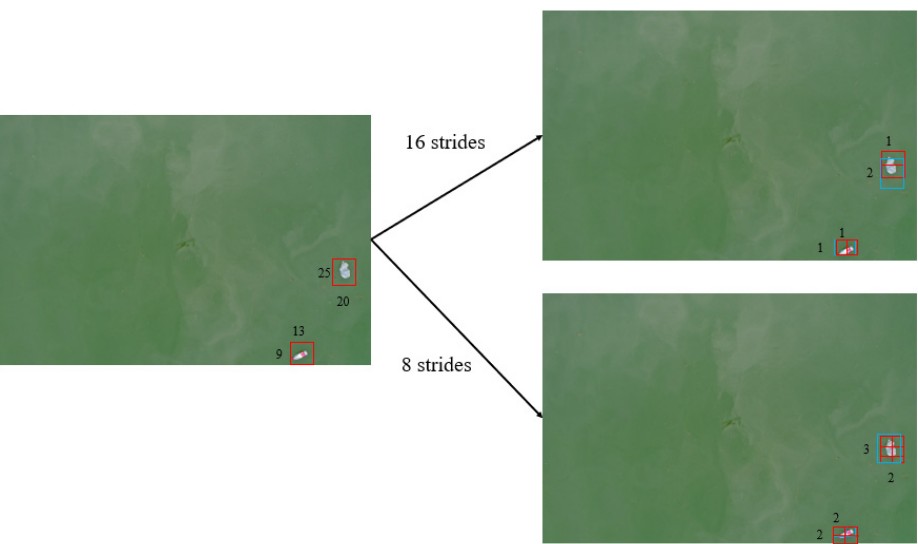

**Figure 7.** Step comparison chart.

3.     Mechanism of increasing attention

The spatial information of river plastic waste is changeable, and the target is difficult to detect. To detect river plastic waste, an attention mechanism was introduced efficiently and accurately into the model, thereby ignoring irrelevant information and stressing localized effective information. Common attention mechanism modules consist of the se module and CBAM module, among others. This study introduced the Yolov5 model into the CBAM module. The CBAM module is lightweight, with the structural features presented in Figure 8. Given an intermediate feature graph, this study inferred the attention weight by complying with the two dimensions of space and channel, and then multiplied it with the original feature graph to adaptively regulate the feature. Since CBAM is a lightweight general module, it can be seamlessly integrated to any CNN architecture, and the extra overhead is negligible. Moreover, it can be trained end-to-end with basic CNN, and

the results can make the model more sensitive to channel and spatial features, and the performance can be enhanced with a small amount of computation.

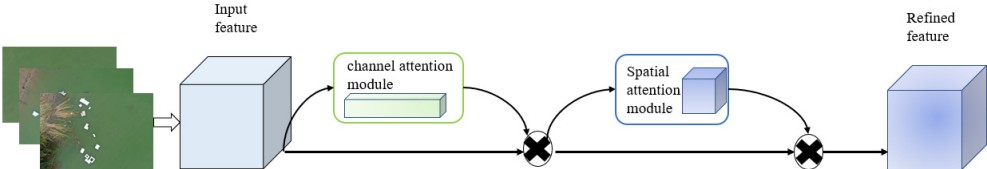

**Figure 8.** CBAM structure diagram.

## 3. Results

### 3.1. Experimental Platform

The airborne image processing board applied experimentally was a TX2 embedded platform for unmanned intelligent field launched by NVIDIA company (Figure 4). It was a modular AI supercomputer, with the GPU of NVIDIA Pascal™ Architecture with 256 CUDA cores. Its CPU covers six cores, consisting of a dual-core denver2 processor and a four-core arm cortex-a57. TX2 is powerful in performance and small in shape. It is significantly applicable to intelligent edge equipment (e.g., a robot, a UAV and an intelligent camera). After training the Yolov5 model on a virtual machine, the model was imported into the TX2 processing board via an SD card. Through the corresponding path of the trained model, the processing results were yielded.

### 3.2. Model Training Results

In this study, the Pascal VOC matrix reported by Everingham et al. [29] was used as the evaluation protocol to verify false positives (FP), true positives (TP) and false negatives (FN). When a predicted bounding box corresponds to a unique real bounding box, it is counted as a TP when it has the largest IOU with a specific real bounding box and reaches the IOU threshold (0.8). Otherwise, the predicted bounding box is considered a FP. When the real bounding box cannot be combined with the predicted bounding box when the IOU reaches the IOU threshold (0.8), it is considered a FN. The prediction of plastic waste in our study is evaluated based on recall (R) and precision (P), which are defined as follows:

$$\text{Precision} = \text{TP}/(\text{TP} + \text{FP}) \tag{1}$$

$$\text{Recall} = \text{TP}/(\text{TP} + \text{FN}) \tag{2}$$

Recall provides insight into the predicted coverage of plastic floating waste, while accuracy evaluates the accuracy of the predicted total. Since the recall rate and precision only reflect one aspect of the model's performance, the average precision (AP) and F1 score were used to comprehensively evaluate the results. AP can be simply regarded as the area under the accurate recall curve or expressed mathematically as:

$$\text{AP} = \Sigma_{i=1}^{n}\text{Precision}_i(\text{Recall}_i - \text{Recall}_{i-1}), \text{withRecall}_{i=0} = 0 \tag{3}$$

The average accuracy of the whole class represents the average value of the whole class mAP, it shows the ability of the target detection model to distinguish different floating plastic wastes.

$$\text{mAP} = \Sigma_{i=1}^{n}\text{AP}/n \tag{4}$$

The score threshold of the algorithm was set to 0.8 to suppress low score prediction. High score predictions were compared with surface facts to yield a set of TP, FP, FN, precision, recall and AP, mAP.

Deep learning models (i.e., Fast-RCNN, YOLOv5 and modified YOLOv5) were adopted to train three types of plastic waste datasets (i.e., plastic bottles, plastic bags and foam

plastics), respectively, on GTX 1080. The data were annotated through 100 iterations of 100 models, and the Yolov5 model P, R, AP and mAP curves were modified (Figure 9).

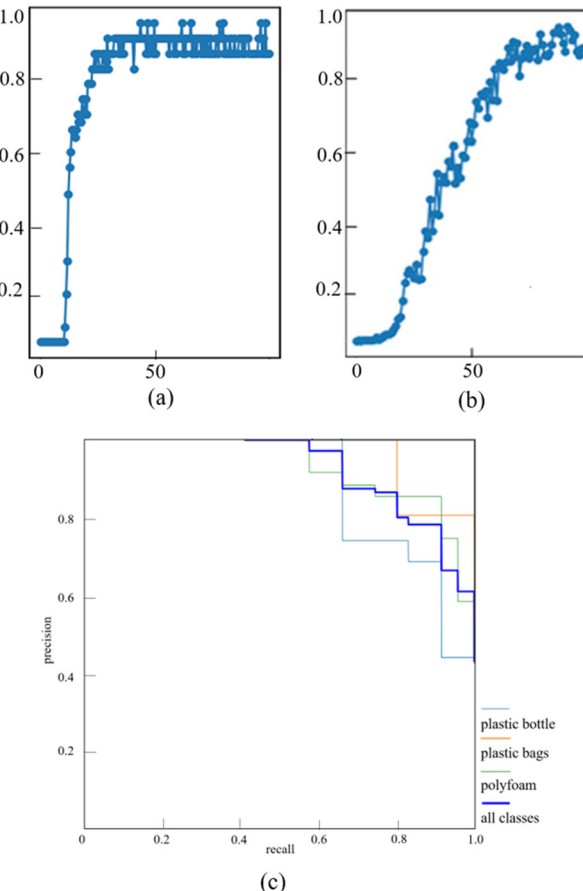

**Figure 9.** The modified YOLOv5 accuracy evaluation. (**a**) Precision curve; (**b**) recall curve; (**c**) AP, mAP curve.

The P curve has converged in the iteration to about 50 generations, and the R curve converged in the iteration to 75 generations. The accuracy of AP and mAP remained more than 80%, which shows that the model converged and the detection effect was good.

*a.    Performance Evaluation*

1.    Recognition results

Given the training results, three types of marine plastic waste were detected, and the results are illustrated in Figure 10.

Because the garbage target was smaller, the local part was enlarged, the yellow border was detected as a plastic bottle, green was plastic foam, purple was plastic bag, the three plastic waste scores were all above 0.8 and the detection results were more accurate.

2.    Accuracy comparison

All experimental models were migrated to the development board for experimental comparison. As assisted by FPS (frames per second), AP, mAP and size of model, a comparison was drawn for the accuracy of the detection results generated by Fast RCNN, SDD, Yolov3, Yolov5 and the modified Yolov5 model (Table 1). After the modified Yolov5 model was transferred to the TX2 development board, FPS could reach 45, and mAP was 94.55%, which indicated optimal overall performance. Thus, the requirements of users to obtain the analysis results of the deep learning model in real time could be more effectively met.

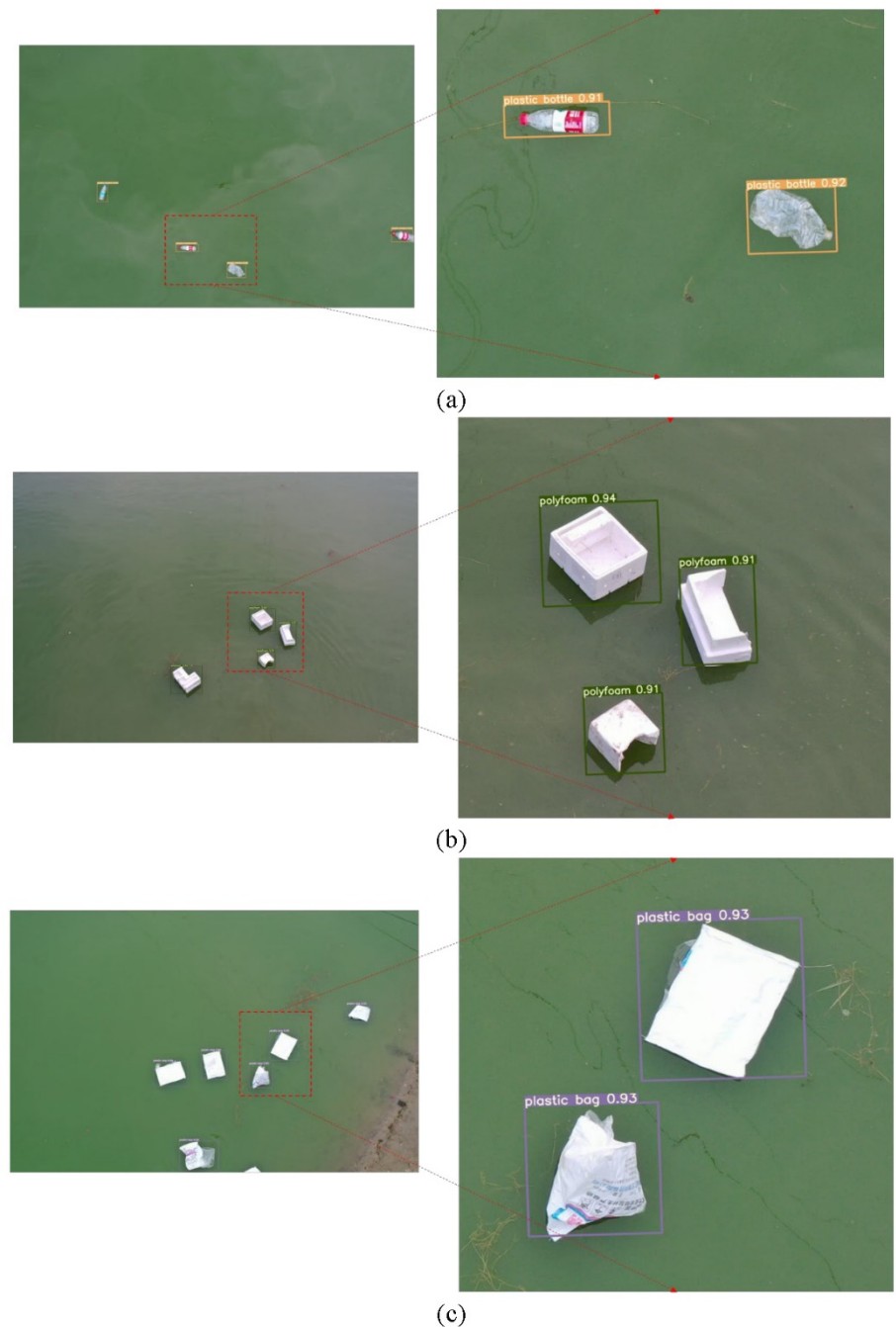

**Figure 10.** Marine litter monitoring results. (**a**) Plastic bottles; (**b**) polyfoam; (**c**) plastic bags.

**Table 1.** Detection results of the dataset.

| Network | FPS | P | R | $AP_{bottle}$ | $AP_{bag}$ | $AP_{polyfoam}$ | mAP | Size of Model |
|---|---|---|---|---|---|---|---|---|
| Faster-CNN | 10.24 | 0.88 | 0.87 | 0.82 | 0.88 | 0.86 | 0.85 | 345 MB |
| SSD | 26.24 | 0.69 | 0.65 | 0.62 | 0.68 | 0.65 | 0.65 | 35.6 MB |
| Yolov3 | 35.62 | 0.73 | 0.76 | 0.71 | 0.73 | 0.75 | 0.73 | 236 MB |
| Yolov5 | 46.37 | 0.80 | 0.82 | 0.81 | 0.82 | 0.80 | 0.81 | 14.5 MB |
| Modified Yolov5 | 43.63 | 0.86 | 0.89 | 0.80 | 0.89 | 0.87 | 0.86 | 15.2 MB |

## 4. Discussion

The Yolov5 network acts as a prominent lightweight network structure. The Yolov5 network's far better performance is primarily attributed to focus structure slice pictures,

extract features of cspnet and optimization strategies of giou using output. Although the fast RCNN detection accuracy was equated with that of the optimized Yolov5, FPS was only about 10 due to the calculation burden attributed to its two-stage structure, so it could not easily meet the real-time requirements of data analysis. By modifying the original Yolov5, the three optimization strategies had different degrees and exerted different effects on the plastic detection classification. The adjustment of the anchor frame and the step length of the anchor frame primarily aimed to address the problem of difficult recognition of small target river waste by adopting the K-means clustering algorithm. The algorithm placed its focus on the plastic waste target in the UAV image more efficiently by introducing an attention detection layer while integrating the information of the plastic waste. Although the optimized Yolov5 increased the model parameters and calculation amount, only 3 FPS were lost, whereas the model's accuracy was elevated by nearly 5%, thereby satisfying the real-time performance and improving the reliability of the results. Furthermore, as verified by the experimentally achieved results, the optimized Yolov5 outperformed other detection algorithms.

The high-precision detection model increased the accuracy in the detection of marine plastic waste, though some plastic waste was not detected due to the influence of the photo environment and angle. On the whole, the recognition accuracy of plastic foam and plastic bags was high, and the recognition accuracy of plastic bottles was slightly lower. The former primarily resulted from the single recognition type and being easy to distinguish, while the latter was largely attributed to the large number of shapes, colors and types, which limited the recognition accuracy. Accordingly, the model could be further optimized by increasing the scale of the dataset and collecting data from different environments.

As indicated from the experimentally achieved results, modified Yolov5 outperformed other target detection methods. High-speed garbage detection can process images in real time and offer floating garbage information for UAV in time in a changeable and complex water environment. Although the fast RCNN achieves high accuracy, it cannot achieve real-time performance due to the computational burden of a two-stage network, and the model size is 345 mb. The huge model hinders the deployment of a UAV algorithm. Yolov5 adopts the focus structure to slice the image, thereby improving the model detection speed without image information loss. The model size is 14.5 mb, meeting the real-time requirements. High-precision plastic detection can help the UAV platform complete the task more accurately, reliably and stably. SSD and Yolov3 are far less accurate than the Yolov5 network. The backbone feature extraction network exerts a certain effect on the performance of the target detection model. The backbone feature extraction network of SSD refers to the classic vgg16 and res101 network, the backbone feature extraction network of Yolov3 is Darknet, and the backbone feature extraction network of Yolov5 is BottleNetCSP. The performance of BottleNetCSP is noticeably better than that of conventional vgg16, res101 and other networks. Its performance is equated with that of the Darknet classifier, yet it has fewer floating-point operations and faster speed, thereby satisfying the practical needs of speed and accuracy.

Yolov5 is prominent in accuracy and speed, so the rising space will not be significant when improving. Accordingly, our goal is to achieve high speed while at least improving the original accuracy. The anchor box reclustered by the K-means clustering algorithm will be more applicable to three types of micro water surface garbage monitoring. Reducing the step size can make the Yolov5 model find water surface plastic garbage better. Introducing a CBAM attention detection layer enabled the target detection algorithm to focus on the plastic garbage targets in UAV images more quickly. The three strategies improved the accuracy of Yolov5. The optimized Yolov5 increased the model parameters and complicated the calculation, and the model size was 15.2 mb; the FPS lost was only 3, whereas the model accuracy increased by about 5%. It was also verified that the performance of the optimized Yolov5 was better than that of other detection algorithms.

Some plastic wastes were not detected by the modified Yolov5 due to the influence of photographing environment and angle. On the whole, the detection accuracy of plastic foam

and plastic bag was high, and the recognition accuracy of plastic bottles was slightly lower. The former was mainly due to the single recognition type and being easy to distinguish, while the latter was primarily attributed to more shapes, colors and types, which limited the recognition accuracy. Accordingly, the model could be further optimized by increasing the scale of the dataset and collecting data from different environments.

The Yolov5 network comprises different sizes of four architectures (i.e., Yolov5s, Yolov5m, Yolov5l and Yolov5x). Users can select specific models with appropriate sizes for development and application. In the present study, the selection and design of the recognition algorithm largely considered its application environment in surface garbage, i.e., the application deployment of detection algorithm on UAV, as an attempt to recognize plastic floating garbage targets in real time. The advantages of light weight (very small model size) and high detection speed of the Yolov5s network will downregulate the deployment cost of the detection model, which shows that the detection model based on the optimized Yolov5s has great potential to be deployed in the edge computing equipment of UAV, and the algorithm can be built by training and strategy selection by complying with different task requirements.

## 5. Conclusions

In this study, the optimized Yolov5 was used to detect three types of common surface plastic waste. As indicated from the experimentally achieved results, the accuracy was further improved compared with that of the original structure though three optimization strategies (i.e., regulating the anchor frame, increasing the detection layer and shortening the step length) that also made the FPS slightly lower. The accuracy of the fast CNN model was equated with that of the Yolov5 model, but the FPS was low and inefficient. In addition, by transferring the trained Yolov5 model to the UAV platform equipped with a TX2 development board, the average accuracy was 86%, and the FPS was 35%. Compared with the existing methods, the method adopted in this study could avoid the significant occupation of network bandwidth attributed to the return video and the lack of information attributed to the analysis delay. In addition, it could effectively achieve the real-time detection and result feedback of all types of water surface plastic waste while significantly increasing monitoring efficiency. The high-intelligence UAV platform is expected to be the mainstream means of water environment monitoring in the future.

**Author Contributions:** Conceptualization and writing—original draft preparation, W.L.; methodology, Z.Z.; software, W.H.; validation, H.W.; formal analysis, M.Z.; investigation, K.L.; data curation, Y.Z.; writing—review and editing, G.W.; visualization, R.X.; supervision, Y.L.; funding acquisition, P.F. All authors have read and agreed to the published version of the manuscript.

**Funding:** This research was funded by National Key Research and Development Program of China, No. 2017YFC0506501; Strategic Priority Science and Technology Special Project of Chinese Academy of Sciences, No. XDA23100203; Key Laboratory of Surveying and Mapping Science and Geospatial Information Technology of Ministry of Natural Resources Open Research Fund Project (2020-2-5); Scientific Research Key Project of Hebei Provincial Department of Education (Grant No. ZD2020161) and Science and Technology Project of Hebei Education Department (QN2019213).

**Institutional Review Board Statement:** Not applicable.

**Informed Consent Statement:** Not applicable.

**Data Availability Statement:** Data for this research are available in ScienceDatabank (doi:10.11922/sciencedb.01121).

**Acknowledgments:** This research was supported by Beijing Droneyee Intelligent Technology Co., Ltd.

**Conflicts of Interest:** The authors declare no conflict of interest.

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
