# Peer review of "A Water Surface Contaminants Monitoring Method Based on Airborne Depth Reasoning"

_processes, doi:10.3390/pr10010131_

Round 1

Reviewer 1 Report

  1. Please, check the article very carefully, as there are typing errors as missing spaces. e.g., not the same line space was used in the first paragraph of the introduction as in the whole article.
  2. Before entering abbreviations, use the whole phrase (abbreviation in parentheses) in the sentence, and only then use abbreviations. Do not use abbreviations in the abstract, headings, or subheadings. g., in the abstract in line no. 19, you used the abbreviation
  3. The keywords need to be in alphabetical order.
  4. I recommended checking the references list because there are some mistakes, and the references style is not the same in the whole list. 
  5. After the Figures and Tables, you should start a new paragraph. The Figure captions on a single line should be centered, be in accordance with the template, and be on the same page. 
  6. The tables should be in accordance with the template.
  7. On page no. 4 (section Materials and Methods), you forgot to delete the text from the template. 
  8. Please, do not use Wikipedia as a reference or source (as on page 3). It should not be a relevant source. 
  9. In section 4 Deep Network, you state that „ Two CSP structures were applied in the backbone extraction network.“ Please, explain the CSP structures briefly. 
  10. In section 2 Model Training Results, you defined four different formulas. Please, describe them in more detail. The same applies to figures 11 and 12. 

Author Response

  1. Please, check the article very carefully, as there are typing errors as missing spaces. e.g., not the same line space was used in the first paragraph of the introduction as in the whole article.

A:  The line spacing of the first paragraph has been adjusted.

  1. Before entering abbreviations, use the whole phrase (abbreviation in parentheses) in the sentence, and only then use abbreviations. Do not use abbreviations in the abstract, headings, or subheadings. g., in the abstract in line no. 19, you used the abbreviation

A:  The abbreviation in line 19 has been changed to the full name.

  1. The keywords need to be in alphabetical order.

A:   Keywords have been arranged alphabetically.

  1. I recommended checking the references list because there are some mistakes, and the references style is not the same in the whole list. 

A:   Revised all the format of all references.

  1. After the Figures and Tables, you should start a new paragraph. The Figure captions on a single line should be centered, be in accordance with the template, and be on the same page. 

A:   The corresponding modifications have been completed as recommended.

  1. The tables should be in accordance with the template.

A:   The corresponding modifications have been completed as recommended.

  1. On page no. 4 (section Materials and Methods), you forgot to delete the text from the template. 

A:   Deleted.

  1. Please, do not use Wikipedia as a reference or source (as on page 3). It should not be a relevant source. 

A:   Deleted.

  1. In section 4 Deep Network, you state that „ Two CSP structures were applied in the backbone extraction network.“ Please, explain the CSP structures briefly. 

A:   Add an explanation of the CSP structure on lines 255 to 262.

  1. In section 2 Model Training Results, you defined four different formulas. Please, describe them in more detail. The same applies to figures 11 and 12. 

A:   Lines 356 to 375 give a more detailed description of the four different formulas.

      Lines 384 to 386 give a more detailed description of figures 11.

      Lines 394 to 396 give a more detailed description of figures 12.

Reviewer 2 Report

General impression#Processes-1531615:

This research is focused on the plastic waste monitoring (from surface waters) by the usage of high-intelligence systems. The best aspect of this engineering approach (sensing systems) presents a wide practical application in the field of environmental prevention and protection. The general impression of the manuscript is that is well structured and written in a style which is clear and understandable. The specific goals and key messages are formulated in a concise manner, based on the results and supported by the previous (literature) studies.

Therefore, the manuscript should be accepted for the publication in the Processes after the several points to be previously addressed:

Comment#1. It is strongly suggested that abbreviations should not be used in the Abstract. Please write the full name and then the abbreviation – Unmanned Aerial Vehicle (UAV). After the abbreviation is introduced, it may be further used through the whole manuscript.

Comment#2. The title is ‘eye catchy’, but syntagma ‘airborne depth reasoning’ was never used again into the whole manuscript. Please, at least, provide this connection to the readers into the Introduction part

Comment#3. Page 1, lines 36-52. The line spacing varies from the rest of the manuscript, please uniform this according to the Guideline for the authors of the Processes.

Comment#4. Page 3, line 103-105. The plastic waste classification (plastic bottles, plastic buckets and plastic straws) is set up by the authors (as the most common types of waste) or it is listed in some of the national regulations – please clarify.

Comment#5. There are too many figures (in total 12) presented in the paper. Although their relevance may be justified, some figures may be merged (in order to improve the flow of the textual interpretation).

Comment#6. Page 4, Materials and Methods, lines 152-166 should be deleted. This text is ‘imported’ from the instructions for the authors. Please correct this omission.

Comment#7. Page 11, Section 3.2 Model Training Results. Please first provide the full name and then the abbreviation (for exp, FP, TP, FN). There are two many abbreviations used in the text; it is recommended to provide the List of abbreviations at the beginning of the manuscript.

Comment#8. Page 13, Line 407. Section 2. Accuracy comparison. There are some technical mistakes – please make a space between ‘SDD, Yolov3, Yolov5 …’

Comment#9. Please uniform the style of the references according to the instructions for the authors (ref 14, 18, 20, 21…are not correctly cited – pagination is missing, some of the references are listed with doi number, while the most of them are not, etc).

Author Response

Comment#1. It is strongly suggested that abbreviations should not be used in the Abstract. Please write the full name and then the abbreviation – Unmanned Aerial Vehicle (UAV). After the abbreviation is introduced, it may be further used through the whole manuscript.

A: The abbreviation in line 19 has been changed to the full name.

Comment#2. The title is ‘eye catchy’, but syntagma ‘airborne depth reasoning’ was never used again into the whole manuscript. Please, at least, provide this connection to the readers into the Introduction part

A: The connection has been added to introduction in line 29-30.

Comment#3. Page 1, lines 36-52. The line spacing varies from the rest of the manuscript, please uniform this according to the Guideline for the authors of the Processes.

A:  The line spacing of the first paragraph has been adjusted.

Comment#4. Page 3, line 103-105. The plastic waste classification (plastic bottles, plastic buckets and plastic straws) is set up by the authors (as the most common types of waste) or it is listed in some of the national regulations – please clarify.

A: The plastic waste classification (plastic bottles, plastic buckets and plastic straws) is set up by the authors.

Comment#5. There are too many figures (in total 12) presented in the paper. Although their relevance may be justified, some figures may be merged (in order to improve the flow of the textual interpretation).

A: We removed Figure 6, because the architecture diagram of Yolov5 can be find in line 248(https://github.com/ultralytics/YOLOv5), and we removed Fig.10, because NVIDIA Jetson TX2 Embedded Platform can be find in Fig.4.

Comment#6. Page 4, Materials and Methods, lines 152-166 should be deleted. This text is ‘imported’ from the instructions for the authors. Please correct this omission.

A:   Deleted.

Comment#7. Page 11, Section 3.2 Model Training Results. Please first provide the full name and then the abbreviation (for exp, FP, TP, FN). There are two many abbreviations used in the text; it is recommended to provide the List of abbreviations at the beginning of the manuscript.

A:   Lines 356 to 375 give a more detailed description of the four different formulas.

Comment#8. Page 13, Line 407. Section 2. Accuracy comparison. There are some technical mistakes – please make a space between ‘SDD, Yolov3, Yolov5 …’

A: Revised.

Comment#9. Please uniform the style of the references according to the instructions for the authors (ref 14, 18, 20, 21…are not correctly cited – pagination is missing, some of the references are listed with doi number, while the most of them are not, etc).

A: Revised and highlight with yellow section.
